# Opportunities and Risks of Promoting Skin and Bone Healing via Implant Biofunctionalization of Extracellular Matrix Protein ECM1

**DOI:** 10.3390/jfb16100385

**Published:** 2025-10-14

**Authors:** Niklas R. Braun, Andreas K. Nüssler, Sabrina Ehnert

**Affiliations:** Siegfried Weller Research Institute, BG Unfallklinik Tübingen, Department of Trauma and Reconstructive Surgery, University of Tübingen, Schnarrenbergstr. 95, 72076 Tübingen, Germany; nbraun2@bgu-tuebingen.de (N.R.B.); sabrina.ehnert@med.uni-tuebingen.de (S.E.)

**Keywords:** wound healing, chronic wounds, non-unions, TGF-β, ECM1, angiogenesis, extracellular matrix

## Abstract

Impaired bone regeneration and wound healing represent a major clinical and socioeconomic challenge for our aging and multimorbid population. Fracture and wound healing share many common features, with transforming growth factor beta (TGF-β) being a key regulator of inflammation, angiogenesis, fibroblast activation, and matrix remodeling. The dysregulation of TGF-β signaling is a hallmark of chronic wounds, excessive scar formation, and fracture non-union. Extracellular matrix protein 1 (ECM1) plays a crucial role in the activation of latent TGF-β. As a protein of the extracellular matrix, ECM1 offers ideal conditions for the biofunctionalization of bone implants or wound patches. Its mode of action has been studied mainly in fibrosis models of the liver or heart, where TGF-β acts as a driver of the disease. The controlled knock-out or overexpression of ECM1 either promoted or improved fibrosis development. In this review, we discuss how these findings can be applied to the biofunctionalization of implants to support bone and wound healing, considering the impact of TGF-β on the different healing phases.

## 1. The Clinical Burden of Fractures, Chronic Wounds, and Dysregulated Healing

Impaired bone regeneration and wound healing represent major clinical and socioeconomic challenges, particularly within aging and multimorbid populations. Fractures and immobility-related pressure ulcers impose substantial healthcare burdens worldwide [1,2,3,4]. Despite the increasing knowledge about the cellular and molecular mechanisms of the underlying healing phases, 2–10% of all fractures result in delayed healing or the formation of non-unions [5,6]. Proximal femoral fractures, complicated by chronic ulceration, are common scenarios linking bone healing and soft tissue repair [7,8,9]. Chronic wounds affect up to 2% of the elderly population and are associated with high mortality, a loss of independence, and spiraling healthcare costs [10,11,12]. The failure of proper healing leads to considerable pain and the functional impairment of the affected extremity, reducing the quality of life for the affected patient. Moreover, protracted rehabilitation, as well as the loss of labor and productivity, represent a significant economic burden on our healthcare system [13,14]. Therefore, elucidating the molecular mechanisms underlying wound and fracture healing remains a critical area of research.

Central to the healing process is the extracellular matrix (ECM), a dynamic and complex microenvironment that orchestrates cell migration, angiogenesis, fibroblast activation, and scar formation [15,16,17]. Growth factor–ECM interactions, particularly involving transforming growth factor-β (TGF-β), play a decisive role in modulating the balance between regenerative repair and fibrotic scar formation [18,19,20,21].

Wound and fracture healing are coordinated in sequential but overlapping phases: hemostasis, inflammation, proliferation (granulation tissue, angiogenesis, and fibroblast activity), and remodeling/maturation [22,23,24], of which hemostasis, inflammation, and early proliferation are comparable in wound and fracture healing (Figure 1). TGF-β is central in orchestrating this process.

### 1.1. Hemostasis

Soft and hard tissue injuries usually involve the disruption of blood vessels. To stop the bleeding, a coagulation cascade is initiated within minutes after injury, which requires the activation of platelets. During this process, platelets release large amounts of active TGF-β (predominantly TGF-β1) from their alpha granules. The resulting local increase in active TGF-β helps to attract immune cells to the site of tissue damage and control their function in the following inflammatory phase [25,26].

### 1.2. Inflammation Phase

Within hours after injury, neutrophils are the first immune cells to invade the wound or fracture. They also release active TGF-β (predominantly TGF-β3) from their granules [27]. Typically, within 1 to 3 days after injury monocytes invade the damaged tissue. The local increase in active TGF-β within the wound or fracture then modulates macrophage phenotypes to resolve the inflammation [25,26]. First, pro-inflammatory M1 macrophages aid in clearing the wounded tissue from cellular debris and invading pathogens; after this, pro-fibrotic M2 macrophages are involved in the resolution of inflammation. This also includes the modulation of immune cells from the adaptive immune system that invade the wounded tissue in the following days and weeks.

### 1.3. Proliferation Phase

Usually after 2–5 days the proliferation phase starts. In the following course of healing, less and less active TGF-β is provided by immune cells. Instead, TGF-β needs to be activated from the extracellular reservoir of latent TGF-β by proteolytic cleavage from latent TGF-β binding protein (LTBP) or by mechanical forces or the acidification of pH. In the proliferation phase, TGF-β stimulates fibroblast differentiation into myofibroblasts [28], promotes angiogenesis [29,30], and regulates ECM deposition [18]. While the regulation of angiogenesis (endothelial cells) and the recruitment of tissue fibroblasts is comparable in wound and fracture healing, the additional invading cells and the formed ECM differ.

#### 1.3.1. Proliferation Phase in Wound Healing

In wound healing, keratinocytes together with fibroblasts form the granulation tissue, mainly composed of pro-collagen, elastin, proteoglycans, and hyaluronic acid. The granulation tissue allows the ingrowth of new blood vessels (formed by endothelial cells) that provide nutrition and oxygen to the growing tissue. Keratinocytes and fibroblasts further induce wound contraction to support wound closure by epithelial cells (re-epithelialization). The proliferating keratinocytes thus restore the barrier function of skin by forming an immature scar [31,32].

#### 1.3.2. Proliferation Phase in Fracture Healing

In fracture healing, invading mesenchymal progenitor cells proliferate and differentiate to chondrocytes and osteoblasts in order to form first the soft and later the hard callus tissue. The soft callus is formed within the first weeks after fracture. Mesenchymal progenitor cells differentiate into chondrocytes to produce a temporary and flexible cartilaginous ECM. This so-called soft callus stabilizes the fracture; however, it is not yet strong enough to bear weight. Typically, within 6–12 weeks after injury, the soft callus is replaced by hard callus. During this process, the formed chondrocytes undergo hypertrophy and osteoblasts deposit minerals like calcium and phosphate into the soft callus. The resulting mineralization gradually strengthens the ECM and thus enables loading. The hard callus represents immature woven bone that is mainly composed of type I collagen and calcium phosphate [33].

### 1.4. Remodeling Phase

The remodeling phase aims to restore the pre-injury state of the skin or bone tissue. In this step, balanced TGF-β activity ensures proper ECM turnover via orchestrating collagen deposition and metalloproteinase activity [34,35].

#### 1.4.1. Remodeling Phase in Wound Healing

The remodeling of wound tissue can last up to a year and is mainly controlled by fibroblasts, myofibroblasts, and macrophages. During this process, the ECM molecules that were produced in a disorganized manner during the proliferative phase are realigned and cross-linked. This process leads to a gradual contraction of the tissue over time. This leads to regaining tissue integrity after wound healing, typically resulting in the formation of a scar, with reduced tensile strength compared with the surrounding skin [31,32].

#### 1.4.2. Remodeling Phase in Fracture Healing

The remodeling of fracture tissue may last up to 2 years and involves osteoblasts and osteoclasts for balanced bone formation and resorption, respectively. During this process the immature woven bone (hard callus) is replaced with mature lamellar bone, in order to regain original bone strength and structure without the formation of scar tissue. More precisely, the center of the callus is replaced by compact bone and the callus margins are replaced by lamellar bone. Alongside these changes, significant remodeling of the vascular system occurs [33].

While reduced TGF-β signaling is associated with impaired bone and wound healing [36,37,38,39] its prolonged or enhanced signaling is associated with fibrotic conditions, keloids, hypertrophic scar formation, and chronic wounds [19,35,40]. Thus, the regulation of TGF-β is essential to steer repair toward regeneration rather than fibrosis.

The effects of TGF-β on, as well as its interaction with, classical ECM constituents such as collagen, elastin, and fibronectin have been extensively studied in the past., However, recent evidence highlights extracellular matrix protein 1 (ECM1) as a crucial, yet underrecognized, regulator of tissue regeneration and repair.

This review aims to elucidate ECM1’s potential role in wound and fracture healing through its modulation of TGF-β signaling involved in the regulation of inflammation, angiogenesis, and new tissue formation/remodeling. Furthermore, this review aims to propose different strategies for how ECM1 can be utilized to aid wound and fracture healing, given specific pathologies with altered TGF-β signaling.

## 2. Regulatory Role of Extracellular Matrix Protein 1 (ECM1)

### 2.1. ECM1: A Multifunctional ECM Protein in Tissue Repair

Extracellular matrix protein 1 (ECM1) is a glycoprotein secreted into the ECM. Up to now, two splice variants for ECM1 have been described in humans, ECM1a and ECM1b. ECM1a, encoded by a 1.8 kb mRNA, is mainly expressed in the highly vascularized heart and placenta, but is also present in the liver, ovaries, kidneys, lungs, pancreas, testes, muscle, and colon. ECM1b, encoded by a 1.4 kb mRNA lacking exon 7, is expressed in the skin and tonsils [41]. In these tissues, ECM1 is predominantly expressed by keratinocytes, fibroblasts, and endothelial cells [42]. It regulates skin barrier integrity, angiogenesis, keratinocyte differentiation, and matrix organization [43]. Its clinical relevance is underscored by Lipoid Proteinosis, a genetic disease caused by ECM1 mutations, characterized by scarring and impaired skin integrity [44].

In wound healing, ECM1 has been shown to (i) modulate angiogenesis by interacting with VEGF and perivascular ECM components [17,45]; (ii) bind to perlecan and influence basement membrane assembly and keratinocyte migration [29,46]; (iii) regulate matrix turnover by inhibiting MMP-9 activity, thereby balancing ECM degradation and deposition [17,47].

Regarding bone regeneration and fracture healing, ECM1 has been shown to (i) be expressed in differentiating chondrocytes upon the activation of parathyroid hormone-related peptide signaling [48]; (ii) negatively regulate chondrogenesis and endochondral ossification both in vitro and ex vivo [48,49]; (iii) inhibit skeletal growth when overexpressed in chondrocytes and osteoblasts [48].

In tissue regeneration, ECM1 functions as an integral mediator balancing angiogenic activity and fibrotic suppression, both of which are tightly regulated by TGF-β-dependent molecular mechanisms [15,17,46,47,50,51]. Moreover, ECM1 deficiency correlates with abnormal myofibroblast persistence, excessive collagen I deposition, and impaired re-epithelialization—all classically associated with TGF-β dysregulation [35,52].

### 2.2. Interplay of ECM1 with TGF-β Pathway

TGF-β, with its three isoforms (TGF-β1, -β2, and -β3), is by far the most abundant cytokine within the TGF-β superfamily. TGF-β is expressed as a latent protein, the so-called latency-associated peptide (LAP). The LAP is covalently bound to LTBP through disulfide bonds in the endoplasmic reticulum. The resulting large latent complex (LLC) is then incorporated into the extracellular matrix, waiting to be activated either enzymatically by proteolytic cleavage, chemically by a drop in pH, or mechanically by interaction with integrins [53]. The released active TGF-β dimers transduce their signals through two types of serine/threonine kinase receptors, termed type I and type II [54]. The type II receptors are constitutively active kinases, which phosphorylate type I receptors upon ligand binding. Seven type I receptors termed activin receptor-like kinase (Alk)-1 through -7 were identified in mammals, of which TGF-β1-3 preferably bind Alk-4 and Alk-5. Upon activation by the type II receptor, Alks activate (phosphorylate) Smad transcription factors in the cytoplasm. TGF-β1-3-activated signaling is mainly mediated via the receptor-regulated Smads (R-Smads) -2 and -3, which upon phosphorylation complex with the common partner Smad (Co-Smad) 4 to enter the nucleus and regulate the expression of target genes, including genes participating in feedback mechanisms of the signaling pathway itself [36,54]—for overview see Figure 2.

Mechanistically, ECM1 modulates TGF-β signaling through the regulation of the growth factor bioavailability. By binding matrix proteoglycans and interacting with collagen/elastin networks [46,47,50], ECM1 influences how latent TGF-β binding proteins (LTBPs) store and present TGF-β in tissues [51]. There are different modes of actions described [55]. Integrins, such as αvβ3, αvβ5, and αvβ6, can mechanically activate TGF-β by connecting the LLC to the contractile cytoskeleton. This requires binding the integrins to the RGD (arginine–glycine–aspartic acid) motif in the LAP. It has been reported that ECM1 protects TGF-β from activation by competitively binding to this RGD sequence, thus preventing the interaction of LAP with the integrins [55]. ECM1 was further reported to interfere with the proteolytic activation of TGF-β by blunting the activity of matrix metalloproteinases 2 and 9 (MMP2 and MMP9), ADAMTS1 (a disintegrin and metalloproteinase with thrombospondin motifs 1), and TSP1 (thrombospondin 1) by interacting with their intrinsic KTRF (lysine–tryptophan–arginine–phenylalanine) and KRFK (lysine–phenylalanine–arginine–lysine) motifs, respectively [45].

## 3. TGF-β Levels in Physiological and Pathophysiological Wound and Fracture Healing

Physiological wound and fracture healing requires a burst release of active TGF-β from platelets leading to a steep increase in active TGF-β levels at the site of tissue damage during the initial hemostasis [56]. During the inflammatory phase, infiltrating neutrophils then provide more active TGF-β, but with an altered isotype composition, inducing negative feedback mechanisms which result in a continuous decline of active TGF-β levels down to basal levels [27] (Figure 3A).

When the initial burst increase in active TGF-β levels at the site of tissue damage is disrupted, the subsequent healing cascade may not be adequately initiated, leading to delayed or impaired wound and fracture healing [57]. This might have different reasons, such as smoking or chronic inflammation. Smokers were reported to have increased numbers of platelets compared with non-smokers. Furthermore, nicotine possesses the ability to activate platelets [58]. However, smoking has been associated with a suppressed increase in active TGF-β after trauma (1 in Figure 3B) [58]. The situation is different in patients with chronic inflammation, where high levels of active TGF-β are continuously secreted by immune cells. This is mainly observed in obese patients, patients with diabetes mellitus [59,60,61,62,63,64,65], or patients with fibrotic diseases of the liver, kidney, heart, or other tissues [66,67,68]. The resulting elevated basal active TGF-β levels limit the local increase at the site of tissue damage (2 in Figure 3B).

Likewise, when the resolution of inflammation fails, the resulting prolonged activation of TGF-β may lead to increased scar or keloid formation [19], or suppress bone formation by impaired mechanotransduction in osteogenic cells (3 in Figure 3B) [69].

## 4. Proposed ECM1 Effects During Wound and Fracture Healing

Targeting the ECM1 as the extracellular regulator of TGF-β signaling offers potential therapeutic strategies, as ECM1 supplementation via biomaterials or gene therapy could stabilize the wound ECM, facilitate controlled angiogenesis, and restore growth factor presentation. However, the proposed effects strongly depend on the healing phase and the predominant TGF-β isoform involved [21]. While TGF-β1 and TGF-β2 produce similar effects, the effect of TGF-β3 is partly opposite.

Tissue injury typically involves the rupture of blood vessels. The resulting exposure of platelets (thrombocytes) to subendothelial collagen leads to platelet aggregation, degranulation, and the activation of the coagulation cascade. The formed fibrin clot both stops the bleeding and serves as a scaffold for the migration of inflammatory cells into the injured tissue. Upon activation, platelets release the content of their granula. Platelet alpha granules are a particularly rich source of active TGF-β isoforms, especially of active TGF-β1, which is up to 100-fold more abundant than in other cell types (the TGF-β1:TGF-β2:TGF-β3 ratio is 4000:1:10) [56]. This results in a rapid and strong local increase in active TGF-β isoforms at the site of tissue damage early during hemostasis. The released TGF-β1 and TGF-β2 act as potent chemoattractants and inflammatory mediators for various types of immune cells, such as neutrophils, mast cells, and monocytes (Figure 4). In the invading neutrophils, the ratio of TGFβ isoforms is biased towards TGF-β3 (the TGF-β1:TGF-β2:TGF-β3 ratio is 12:1:34) [27], which is partly antagonizing the effects of TGF-β1 and TGF-β2 advancing the healing process towards the proliferation phase (Figure 5). As with hemostasis and the initial inflammation phase, active TGF-β is released by platelets and neutrophils and only very mild or even no antagonizing effects of ECM1 are expected. This is in contrast to the following healing phases, which require the activation of TGF-β from the ECM.

In the proliferation phase three major events are mediated by TGF-β, namely re-epithelialization, angiogenesis, and formation of ECM. All three TGF-β isoforms have been reported to promote re-epithelialization by inducing proliferation and migration of epithelial cells at the wound margins [27,70]. However, with one exception for in vitro experiments, where keratinocyte migration was promoted only by TGF-β1 but not by TGF-β3 [71]. During the following angiogenesis, the endothelial cells form capillary sprouts that invade the wounded tissue to form a de novo microvascular network. The role for TGF-β as a modulator of angiogenesis is strongly context dependent and includes the recruitment of vascular endothelial growth factor (VEGF)-producing hematopoietic effector cells to the site of tissue damage, local induction of VEGF expression [29,72] and induction of endothelial to mesenchymal transition [73]. However, endothelial to mesenchymal transition has also been widely associated with pathological fibrosis of various organs, including the skin [74]. Finally, all three TGF-β isoforms participate in fibroblast recruitment to the site of injury and their activation to produce the provisional ECM; however, this process is strongly isotype-dependent. TGF-β1 is reported to induce collagen production, specifically collagen type I and III. Hence, excessive TGF-β1-mediated signaling has been associated with scarring and the development of keloids [75]. The less abundant TGF-β2 shows similar effects than TGF-β1, unlike TGF-β3 which appears to be anti-fibrotic [76].

This isotype-specific effect is enhanced in the following remodeling phase where TGF-β regulates the transition from fibroblasts to myofibroblasts, a population of fibroblasts with a contractile phenotype. However, this effect is strongly dose-dependent. Myofibroblasts are characterized by the expression of alpha smooth muscle actin (αSMA), which is controlled by TGF-β1, both through SMAD-dependent and independent signaling. Thus, the suppression of TGF-β1 in this phase of healing supports scar-free healing while its induction may favor scar formation by the excessive activation of myofibroblasts [28]. Similarly to TGFβ1, TGFβ2 promotes the transition of fibroblasts to myofibroblasts both in vitro and in vivo (Figure 4). The role of TGF-β3, however, is more complex (Figure 5). It appears to promote myofibroblast transition in vitro but inhibits the same process in vivo [76,77].

Especially in the proliferation and remodeling phases, where TGF-β is activated from the ECM reservoir, a controlled regulation of this process offers multiple options for intervention. ECM1 might be established as an ideal candidate for that, due to its extracellular field of action. Thus, it is feasible that timely controlled ECM1 knock-down or neutralization might favor re-vascularization, angiogenesis, and ECM deposition, resulting in accelerated wound closure or fracture healing, as well as a stronger anchoring of implants in the bone. Meanwhile, later in the remodeling phase, a controlled ECM1 induction might prevent scar or keloid formation. However, these timely effects require specific biofunctionalization strategies, which are discussed in the following paragraph.

## 5. Opportunities and Risks: Biofunctionalization Strategies

The biofunctionalization of implants or hydrogels to induce or neutralize ECM1 effects might be achieved by several strategies, including the application of the recombinant human ECM1 protein directly or tetrapeptides, KTRF and KRFK, by regulating their interaction with the proteases ADAMTS1 and TSP1 [45]. Furthermore, implants might be coated with RNA or DNA products to either induce or suppress ECM1 expression in the surrounding tissues (Figure 6). While RNA-based methods primarily influence gene expression at the post-transcriptional level [78], DNA-based strategies typically affect transcription at the genetic or epigenetic level [79].

The induction of ECM1 expression might be achieved by applying synthetic messenger RNA- (mRNA) or DNA-based strategies. Based on the fact that the biological stability of DNA is longer than RNA, the administration of synthetic mRNA is suitable for the short and temporary induction of gene expression [78]. Importantly, DNA-based gene delivery may alter the genome itself, depending on the method used, while the use of episomes and plasmids (circular DNA) allows for the transient expression of the transferred gene, the use of transposons (“jumping genes”), and CRISPR-Cas9 gene editing, while CRISPR activation (CRISPRa) leads to the integration of the gene into the genome, allowing for the stable expression of the transferred gene [79]. When using viral gene delivery, it strongly depends on the viral system used and whether the delivered gene integrates (e.g., retroviral- or lentiviral-based gene delivery with stable gene expression) or not (e.g., adeno-associated viral gene delivery with transient gene expression) into the host’s genome [80].

The suppression or knock-down of ECM1 expression might be achieved by various RNA-based techniques, such as RNA interference (double-stranded RNAs) using small interfering RNAs (siRNAs) or short hairpin RNAs (shRNAs); single-stranded microRNAs (miRNAs); long non-coding RNAs (lncRNAs) or aptamers (short single-stranded RNA or DNA molecules) [78]; or DNA-based strategies. However, the latter usually applies methods such as CRISPR-Cas9 gene editing or CRISPR interference (CRISPRi).

### 5.1. Proposed Use of Recombinant Human ECM1

Recombinant human ECM1 at an implant surface should prevent excessive TGF-β activation. The early hemostasis and inflammatory phase might not be affected by such a coating, as their active TGF-β is released from platelets and immune cells, such as neutrophils. However, during the proliferation phase a lack of TGF-β activation might hinder re-epithelialization, angiogenesis, and the formation of the provisional ECM, as observed when all three TGF-beta isoforms are neutralized with neutralizing antibodies [81].

In order to exert its biological effect, ECM1 has to be presented within the extracellular space; therefore, surface-coating techniques facilitating non-covalent and covalent protein bonding with the implant surface may be a promising strategy. Non-covalent coating techniques include techniques such as electrophoretic deposition, precipitation, and dip or spin coating. These methods use physical adsorption by van der Waals forces or electrostatic interactions to bind the protein (dissolved in a liquid) to the implant’s surface. Due to the reversible adhesive forces, non-covalently bound proteins may be rapidly released from the implant surface. In contrast, covalent bonding approaches create a stronger and thus more permanent bond between the protein and the implant surface. To facilitate the covalent bonding of proteins with the implant surface, the implant surface has to be chemically modified by creating reactive sites for proteins by silanization, polyethylene glycol (PEG) coating, or via chemicals that allow ultraviolet light (UV)-mediated crosslinking. Biomaterial scaffolds mimicking the ECM or incorporating ECM fragments have been proposed to enhance wound repair outcomes [82,83]. Therefore, indirect implant coating by peptide bonds or mineral coating that allows the timely release of the ECM1 [84,85] would be preferable to direct coating techniques, such as spray or dip coating. In contrast, the late remodeling phase application of ECM1 might be beneficial to prevent scar or keloid formation in respective risk patients. With this, the recombinant human ECM1 might be delivered later in the healing phase incorporated in hydrogel wound patches or as peptide-based nanoparticles [86,87,88]. However, in both cases the presentation of ECM1 within the extracellular space needs to be provided to ensure its biological function.

### 5.2. Proposed Use of Tetrapeptide Sequences Targeting the Interaction of ECM1 with Proteases

The knowledge of specific motifs either in the ECM1 protein or the interacting proteases offers great perspectives for the modulation of ECM1 effects. Much of this knowledge is obtained from murine studies on the effect of TGF-β signaling in liver fibrosis. On the one hand, the hepatocyte-specific knock-out of ECM1 caused latent TGF-β1 activation and spontaneously induced liver fibrosis with rapid mortality [45,89]. On the other hand, the overexpression of ECM1 was able to attenuate liver cirrhosis in mouse models [90,91]. In patients with chronic liver disease (CLD), ECM1 expression is inversely associated with the levels of TSP1, ADAMTS1, MMP-2, MMP-9, and LTGF-β1 activation [45,92].

By investigating the underlying effects, ECM1 was shown to suppress the activity of the proteases TSP1, ADAMTS1, MMP-2, and MMP-9, which are all involved in latent TGF-β1 activation. Immunoprecipitation experiments proved that ECM1 binds directly to these enzymes. In the case of TSP1 and ADAMTS1, specific tetrapeptide sequences were identified that control this interaction—the KRFK and KTRF motifs [45]. Both motifs are proposed to induce a conformational change in the LTGF-β1 molecule, resulting in the release of the active TGF-β1 ligand without the need for additional proteolysis. In line with this, the application of KRFK tetrapeptides, which is characteristic for the interaction with TSP1, induced the activation of latent TGF-β. This effect was partly attenuated by the overexpression of ECM1 [45]. In contrast, the application of KTRF tetrapeptides suppressed the activation of latent TGF-β, even in ECM1 knock-out mice [45].

The knowledge of this opposite effect of these two tetrapeptides offers great perspectives for the biofunctionalization of implants. In particular, due to their very small size, tetrapeptides can be applied to implants in larger amounts than the recombinant human ECM1 by using the same application methods described before. As the KTRF tetrapeptide simulates the ECM1 effect, its application might be beneficial later in the healing cascade when a resolution of the inflammation is required to prevent excessive scar formation. The KRFK tetrapeptides, which were shown to antagonize the ECM1 effects, might exert beneficial effects early in the proliferation phase, such as improving re-epithelialization, angiogenesis, and ECM production.

### 5.3. Proposed Use of RNA- and DNA-Based Methods to Regulate ECM1 Expression

Using RNA- or DNA-based methods might have the advantage that the target gene needs to be expressed and secreted by the local cells, which causes a natural delay in the proposed effects. This is advantageous when ECM1 will be overexpressed, as this is supposed to interfere with the early phases of wound healing. Given this required delay of the proposed effect, this argues for the DNA-based rather than the RNA-based overexpression of ECM1—preferably delivered as episomes or plasmids that allow the transient expression of the transferred gene without integration into the genome.

In contrast, the suppression of ECM1 might be beneficial in the early proliferation phase of the healing cascade, where active TGF-β supports re-epithelialization, angiogenesis, and ECM production. There a timelier effect is required, an RNA-based rather than a DNA-based knock-down of ECM1 can be argued for. However, similar to the DNA-based methods, the delivery technique may have a significant effect on the speed and duration of the response. Usually, lnRNAs and aptamers provide the most rapid response with the shortest duration of the effect lasting between hours and days. This is due to their short half-life, which for lnRNAs is defined mainly by their primary and secondary structures, but also by their interaction with other proteins within the cellular context [93]. The half-life of aptamers, in contrast, is defined by renal clearance and nuclease degradation. However, their half-life and therapeutic effect can be extended through modifications such as PEGylation, which increases their size and resistance to nucleases [94]. An intermediate effect duration (~5 days) is expected from miRNAs, which have an average half-life of about 5 days. Their stability is enhanced when forming a complex with the argonaute proteins as part of the RNA-induced silencing complex (RISC) [95]. Synthetic miRNAs follow the same degradation kinetics as siRNAs; thus, the expected effect duration is also comparable (5 to 7 days). A more persistent effect is obtained by shRNAs that are often expressed from a plasmid or viral vector by the host cells. This might even be permanent, when the delivery method includes integration into the hosts’ genome [96]; however, this is unfavorable for therapeutic use.

## 6. Conclusions

Wound healing depends on tightly regulated TGF-β pathway activity, orchestrated by ECM components, proteases, and receptor modulators. ECM1 emerges as a critical stabilizer of this system, ensuring balanced growth factor availability and matrix organization. As a protein predominantly localized within the ECM, ECM1 is a highly promising candidate for the biofunctionalization of implants due to its principal actions within the extracellular space. In light of its proposed mechanisms, several approaches for biofunctionalization are feasible: Direct strategies include the application of rh-ECM1 or the tetrapeptides, KTFR and KFRK. Indirect strategies may utilize RNA- or DNA-based techniques—such as siRNAs, shRNAs, aptamers, messenger RNA, or plasmids—to achieve the knock-down or the overexpression of ECM1. These molecules can be delivered to implants either alone, with the aid of polymers or vesicles, or incorporated into hydrogels.

Depending on the selected strategy, the delivered protein/peptide, RNA, or DNA may either activate or inhibit the proteases ADAMTS1, MMP2, and MMP9, as well as TSP1, which are involved in the proteolytic release of active TGF-β from its latent complex. Ultimately, leveraging the modulatory capacity of ECM1 in wound and fracture healing—via its regulation of TGF-β signaling—holds significant promise for addressing the substantial clinical challenges posed by non-unions, impaired wound healing, and chronic wounds.

## Figures and Tables

**Figure 1 jfb-16-00385-f001:**
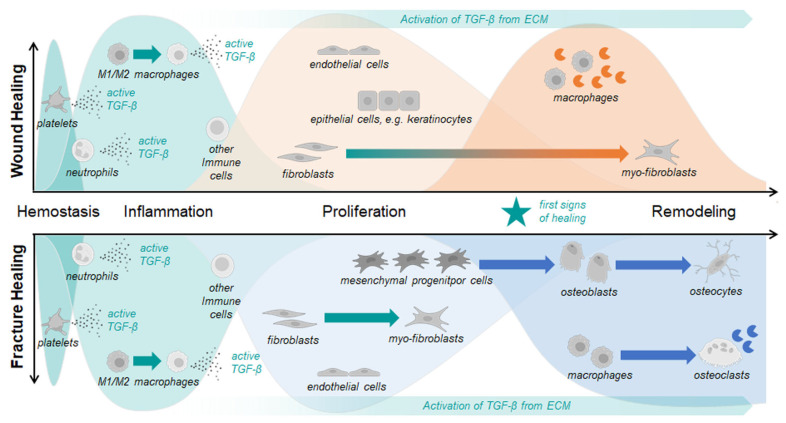
Schematic overview of the different phases of wound and fracture healing with the different cell types involved. Both wound and fracture healing are divided into 4 overlapping phases: (i) hemostasis, (ii) inflammation, (iii) proliferation, (iv) remodeling. While hemostasis and the inflammatory phase are comparable in wound and fracture healing, the later proliferation and remodeling phase are distinguished clearly based on the involved cells. Parts of this figure were Created in BioRender. Rivas, C. (2025) https://BioRender.com/rhhaqwy. To access the original source of the cell images, visit https://app.biorender.com/illustrations/671f5c9f83adf666c3253635 (accessed on 1 October 2025).

**Figure 2 jfb-16-00385-f002:**
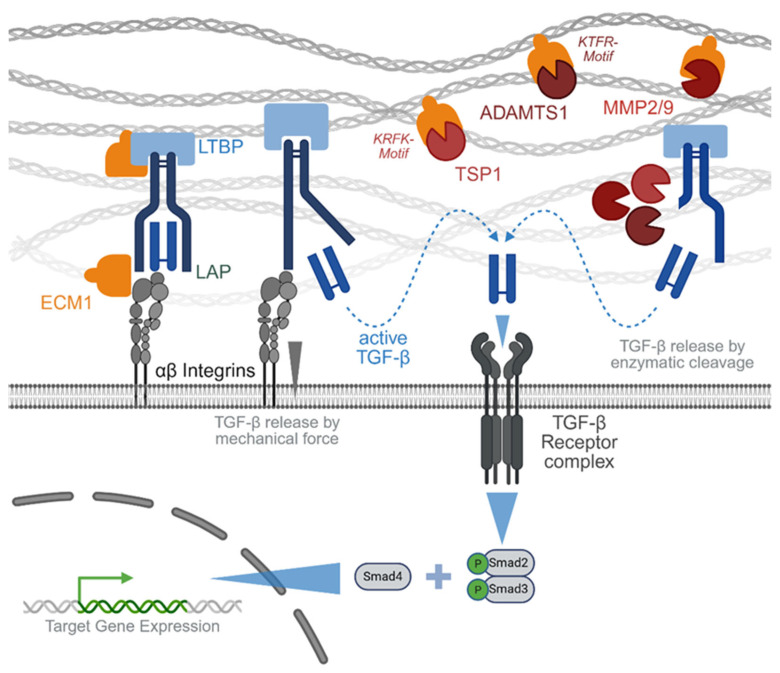
ECM1 as extracellular regulator of TGF-β signaling—proposed modes of action. The extracellular matrix protein 1 (ECM1) is incorporated in the extracellular matrix. There are different modes of action described for how ECM1 may regulate transforming growth factor beta (TGF-β) activation. ECM1 may prevent the release of active TGF-β from the latency-associated peptide (LAP) (i) by strengthening its binding to the latent TGF-β binding protein (LTBP) or (ii) by inhibiting the mechanical opening of LAP by αβ integrins. (iii) Furthermore, ECM1 was reported to bind proteases, e.g., a disintegrin and metalloproteinase with thrombospondin motifs 1 (ADAMTS1—at its KTRF motif), matrix metalloproteinase 2 and 9 (MMP2 and MMP9), and thrombospondin 1 (TSP1—at its KRFK motif), involved in the proteolytic release of active TGF-β from the LAP. Active TGF-β dimers can then bind the TGF-β receptor complex to phosphorylate the intracellular transcription factors’ small mothers against decapentaplegic 2 and 3 (Smad2 and Smad3), which upon binding to Smad4 enter the nucleus to regulate target gene expression. This figure was Created in BioRender. Rivas, C. (2025) https://BioRender.com/rhhaqwy. To access the original source of this figure, visit https://app.biorender.com/illustrations/671f5c9f83adf666c3253635 (accessed on 29 August 2025).

**Figure 3 jfb-16-00385-f003:**
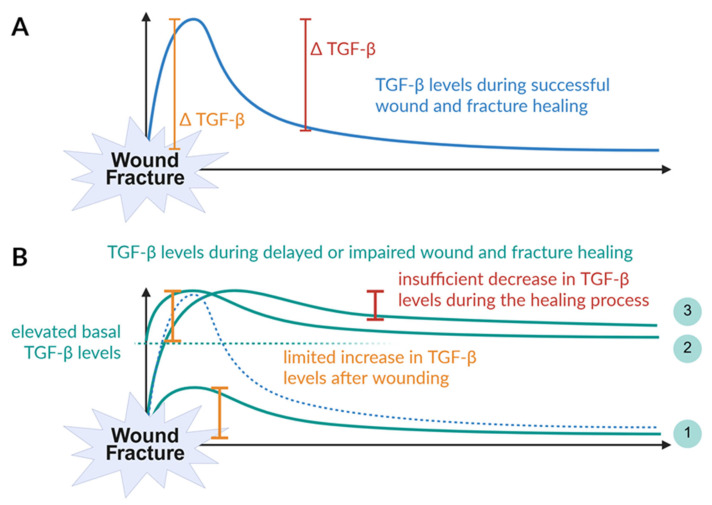
Alterations in active TGF-β levels in wound and fracture healing. (**A**) During physiological wound and fracture healing there is usually a burst release of active TGF-β from platelets leading to a steep increase (Δ) in active TGF-β levels at the site of tissue damage. Infiltrating neutrophils then provide more active TGF-β, but with an altered isotype composition, inducing negative feedback mechanisms which result in a continuous decline of active TGF-β levels down to basal level. (**B**) During delayed or impaired wound and fracture healing, this controlled release of active TGF-β may be disturbed. The healing process may not be initiated when the initial rise in active TGF-β levels is too low, either through inhibition, such as observed in smokers (1) or through already-increased basal levels, such as those observed in patients with chronic inflammation (2). Furthermore, a prolonged activation of TGF-β may lead to increased scar or keloid formation (3). This figure was Created in BioRender. Rivas, C. (2025) https://BioRender.com/rhhaqwy. To access the original source of this figure, visit https://app.biorender.com/illustrations/671f5c9f83adf666c3253635 (accessed on 29 August 2025).

**Figure 4 jfb-16-00385-f004:**
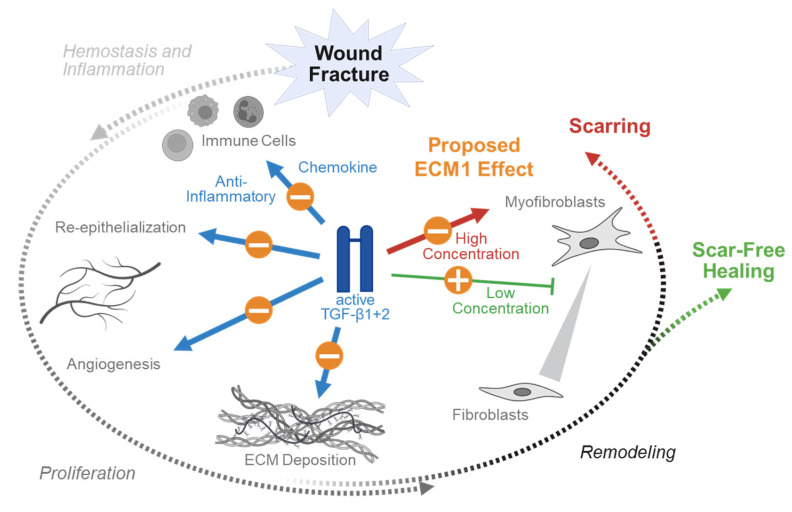
Proposed ECM1 effects during the different phases of wound and fracture healing dependent on its interaction with TGF-β1 and TGF-β2. The three different transforming growth factor beta (TGF-β) isoforms exert specific functions during wound and fracture healing in humans. In the initial phase of hemostasis and inflammation, TGF-β1 and TGF-β2 act as chemokines to recruit immune cells to the wound/fracture. With their anti-inflammatory function they advance the healing process towards the proliferation phase. In the proliferation phase they promote re-epithelialization and angiogenesis, as well as extracellular matrix (ECM) deposition. In the following remodeling phase, TGF-β regulates the transition from fibroblasts to myofibroblasts; however, this effect is strongly dose-dependent. While the suppression of TGF-β1 and TGF-β2 supports scar-free healing, its induction may favor scar formation by the excessive activation of myofibroblasts. The orange circles provide the proposed positive (+) or negative (−) effect of extracellular matrix protein 1 (ECM1), considering that it suppresses the activation of TGF-β1 and TGF-β2. This figure was Created in BioRender. Rivas, C. (2025) https://BioRender.com/rhhaqwy. To access the original source of this figure, visit https://app.biorender.com/illustrations/671f5c9f83adf666c3253635 (accessed on 29 August 2025).

**Figure 5 jfb-16-00385-f005:**
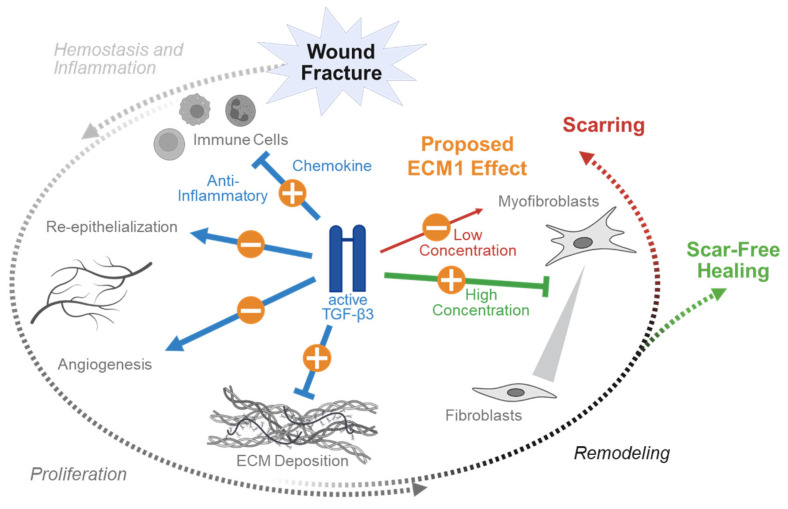
Proposed ECM1 effects during the different phases of wound and fracture healing dependent on its interaction with TGF-β3. Transforming growth factor beta (TGF-β) effects on wound and fracture healing are dependent on the specific TGF-β isoform. In the initial phase of hemostasis and inflammation, the effect of TGF-β 3 is contrary to that of TGF-β1 and TGF-β2. It suppresses the recruitment of immune cells to the wound/fracture. In the proliferation phase TGF-β3 supports re-epithelialization and angiogenesis, but unlike TGF-β1 and TGF-β2 inhibits extracellular matrix (ECM) deposition. In the following remodeling phase TGF-β3 regulates the transition from fibroblasts to myofibroblasts in a dose-dependent manner, where high levels of TGF-β3 are required for scar-free healing. The orange circles provide the proposed positive (+) or negative (−) effect of extracellular matrix protein 1 (ECM1), considering that it suppresses activation of TGF-β. This figure was Created in BioRender. Rivas, C. (2025) https://BioRender.com/rhhaqwy. To access the original source of this figure, visit: https://app.biorender.com/illustrations/671f5c9f83adf666c3253635 (accessed on 29 August 2025).

**Figure 6 jfb-16-00385-f006:**
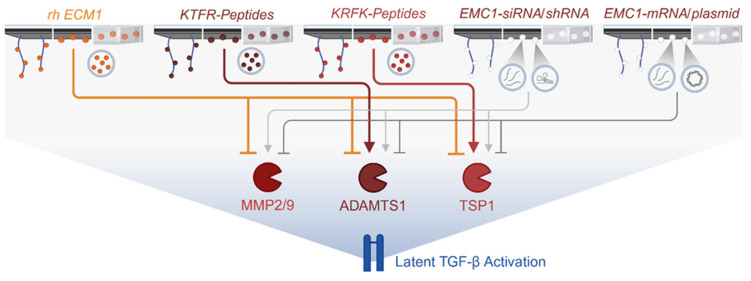
Examples of the biofunctionalization of implants through ECM1. As a protein normally situated in the extracellular matrix, extracellular matrix protein 1 (ECM1) is an ideal candidate for the biofunctionalization of an implant as its mode of action is within the extracellular space. Considering its proposed modes of action, there are different strategies that are conceivable. Directly, by utilizing the recombinant human ECM1 (rh-ECM1) or the tetrapeptides KTFR (lysine–threonine–phenylalanine–arginine) and KFRK (lysine–phenylalanine–arginine–lysine). Or indirectly, using RNA- or DNA-based techniques, e.g., small interfering RNAs (siRNAs), small hairpin RNAs (shRNAs), aptamers, messenger RNA, or plasmids, for either the knock-down or the overexpression of ECM1. These might be applied onto the implants either directly or with the help of polymers or vesicles or integrated into hydrogels. Based on the chosen strategy and the applied protein/peptide, RNA or DNA may cause the activation or inhibition of the proteases, a disintegrin and metalloproteinase with thrombospondin motifs 1 (ADAMTS1), matrix metalloproteinase 2 and 9 (MMP2 and MMP9), and/or thrombospondin 1 (TSP1), involved in the proteolytic release of active TGF-β from the LAP. This figure was Created in BioRender. Rivas, C. (2025) https://BioRender.com/rhhaqwy. To access the original source of this figure, visit https://app.biorender.com/illustrations/671f5c9f83adf666c3253635 (accessed on 29 August 2025).

## Data Availability

No new data were created or analyzed in this study. Data sharing is not applicable to this article.

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
