# Peer review of "Opportunities and Risks of Promoting Skin and Bone Healing via Implant Biofunctionalization of Extracellular Matrix Protein ECM1"

_jfb, 2025, doi:10.3390/jfb16100385_

Round 1

Reviewer 1 Report

Comments and Suggestions for Authors

This manuscript, entitled "Opportunities and risks of biofunctionalization of implants with extracellular matrix protein 1  to support skin and bone healing” addresses the new approach on ECM-1 functionalization for tissue healing. Overall, the topic is pretty interesting with promising potential to be applied in translational research. But, there are some important points should be addressed for better manuscript:

  1. This manuscript seems to be more focused on the role and application of ECM-1 for biofunctionalization on skin and bone healing, meanwhile this is not specified in the title. The authors are highly encouraged to temper the title to be more in line and specified to what they are aiming.
  2. The abstract is quite unclear in the current version, in regard to the aims and health problems the authors trying to solve. The authors need to be very clear in the abstract in order to give better understanding to the readers. It will be very beneficial if the authors could explain the disease burden and aims in the earlier part of the abstract, instead of starting off with ECM-1.
  3. The authors seems to aim to tackle the problem in fractures, and subsequent skin wound healing problems happened to it. But then there are some mention of pressure ulcers and other type of pathologies, thus making this manuscript to be to unfocused. The authors should explain first, their target bone pathologies to tackle, and how this pathologies affects the overlying skin wound, in the introduction. Then the authors could cover the ECM formation in this context. In the current version, the focus is to dispersed and not easy to understand, since the authors explain about bone healing in one sentence, and skin wound in another, without any proper context explanation. The introduction section is also not present in this current version. The first section might be intended to be the introduction section, but it is much better to stick to the proper scientific section names. In the introduction section also, the aim is not supported by proper justification and explanation on why ECM-1 is important to be reviewed in this study. Comparison with previous studies also necessary to be included.
  4. The authors have the tendencies to use non-scientific words and sentences, such as “At the heart of wound healing lies the extracellular matrix…”. This might be not scientifically informative enough and have possibilities for multi-interpretation and misleading.
  5. “During the inflammation phase large amounts of active TGF‑β”. The authors need to explain in detail what condition make the TGF- β to be active from their inactive form, and connect them to the wound healing process.
  6. The 2nd section covers on the signalling process of both wound and fracture healing, but the information written is too brief, general, and superficial, also not classified between two different tissue which have two different pathways. This should be written in detail, so the authors could present in which phase and how the TGF-β might play important part in the healing process. There are not much information that could be extracted from this section in the current version. The same case also for the ECM-1 section (section 3).
  7. Line 68: there is a miswriting of ECM-1 in the subsection title
  8. Line 223: there is a miswriting of biofunctionalization
  9. The authors also encouraged to briefly the protocols and methods that could be used to functionalize protein on implants.

Author Response

We sincerely thank the reviewer for his/her thorough evaluation of our manuscript and constructive suggestions for improvement. We have carefully addressed all points raised, and detailed responses to each question/comment can be found in the attachment.

Reviewer 2 Report

Comments and Suggestions for Authors

The review manuscript focuses on summarizing the effect of ECM-1 on TGF-β signaling regulation in wound and bone fracture healing. It is a well-written manuscript with a broad and logical review structure. The paper covers the major areas related to the relationship between ECM-1 and TGF-β signaling in wound and bone fracture healing. The topic is interesting and may provide valuable insights for readers to further investigate this area. However, the following comments should be addressed before the manuscript can be considered for publication:
1.    The title of Section 5.2 needs to be re-checked.
2.    The topic of Section 5.2.1 is related to RNA- and DNA-based methods to regulate ECM-1 expression. This section may be more appropriately placed under Section 5.1, Protein-, RNA-, and DNA-Based Strategies.
3.    Section 3.1 should include the roles of ECM-1 in bone fracture/defect healing. Is there any difference in the roles of ECM-1 between wound healing and bone fracture/defect healing?
4.    The review aims to cover the effects of ECM-1 in both wound and bone healing. However, wound healing and bone fracture healing follow different pathways. Would it be possible to create two major subsections under Biofunctionalization Strategies—one focusing on wound healing and the other on bone fracture/defect healing?

Author Response

(The authors gave the same response as above.)

Round 2

Reviewer 1 Report

Comments and Suggestions for Authors

In general, the authors have addressed some of the comments provided by the reviewers before this session. But there is one crucial major thing that is yet to be addressed: the author still discusses the healing of soft and hard tissue as one entity together. This is a significant scientific error and misleading, since every tissue has different physiology, and they should be discussed in a focused manner for each. Even if it is to avoid redundancies, please clearly separate and indicate which processes have the same or different events. Please address this first before any recommendation. It will be beneficial if the authors could present this in an illustration. 

Author Response

We would like to thank you for the opportunity to revise once more our manuscript “Opportunities and risks of promoting skin and bone healing via implant biofunctionalization of extracellular matrix protein ECM1” (jfb-3877511). For detailed answers please see attachment.

Reviewer 2 Report

Comments and Suggestions for Authors

The authors have significantly improved the manuscript according to the comments. I have no further comments. 

Author Response

We would like to thank the reviewer for his/her estimate on out manuscript.